# Spatiotemporal dynamics of lesion-induced axonal sprouting and its relation to functional architecture of the cerebellum

Matasha Dhar[1], Joshua M. Brenner[1], Kenji Sakimura[2], Masanobu Kano[3] & Hiroshi Nishiyama[1]

Neurodegenerative lesions induce sprouting of new collaterals from surviving axons, but the extent to which this form of axonal remodelling alters brain functional structure remains unclear. To understand how collateral sprouting proceeds in the adult brain, we imaged post-lesion sprouting of cerebellar climbing fibres (CFs) in mice using *in vivo* time-lapse microscopy. Here we show that newly sprouted CF collaterals innervate multiple Purkinje cells (PCs) over several months, with most innervations emerging at 3–4 weeks post lesion. Simultaneous imaging of cerebellar functional structure reveals that surviving CFs similarly innervate functionally relevant and non-relevant PCs, but have more synaptic area on PCs near the collateral origin than on distant PCs. These results suggest that newly sprouted axon collaterals do not preferentially innervate functionally relevant postsynaptic targets. Nonetheless, the spatial gradient of collateral innervation might help to loosely maintain functional synaptic circuits if functionally relevant neurons are clustered in the lesioned area.

[1] Center for Learning and Memory, Department of Neuroscience, The University of Texas at Austin, 1 University Station Stop C7000, Austin, Texas 78712-0805, USA. [2] Department of Cellular Neurobiology, Brain Research Institute, Niigata University, 1-757 Asahimachidori, Chuo-ku, Niigata 951-8585, Japan. [3] Department of Neurophysiology, Graduate School of Medicine, The University of Tokyo, 7-3-1 Hongo, Bunkyo-ku, Tokyo 113-0033, Japan. Correspondence and requests for materials should be addressed to H.N. (email: hiroshi@utexas.edu).

Axonal degeneration is observed in a wide variety of neurological disorders, as well as following brain lesion. Degenerated axons do not typically regenerate in the brain; instead, surviving axons sprout new collaterals and innervate the territory of lost axons[1–4]. Previous studies suggest that this form of collateral sprouting restores some brain function lost after unilateral brain lesion and delays the progression of Alzheimer's disease towards development of clinical symptoms[2,5–8]. On the other hand, collateral sprouting has also been shown to contribute to the pathogenesis of temporal lobe epilepsy and other neurological disorders[2,9,10]. This two-sided aspect of collateral sprouting on brain function highlights the significance of new axon collaterals regardless of whether they are beneficial or detrimental. Correctly promoting or inhibiting collateral sprouting could prove to be an effective post-lesion therapeutic intervention. A crucial step towards such a future application is to understand when collateral sprouting occurs extensively after lesioning, how much a surviving axon expands its innervation territory, and whether sprouted collaterals selectively innervate functionally relevant postsynaptic neurons.

So far, collateral sprouting has been studied mostly by conventional histological techniques in which dynamic progression of axonal sprouting is only inferred from static images taken from different animals. Therefore, the precise spatiotemporal pattern of collateral sprouting remains largely unclear. In addition, although newly sprouted collaterals roughly follow developmental innervation patterns (that is, innervating the same layers and postsynaptic cell types)[11,12], it is unknown whether new collaterals in the mature brain randomly innervate any available postsynaptic neurons or preferentially innervate neurons within the same functional circuit.

The cerebellum provides a good model to analyse the precise spatiotemporal pattern of lesion-induced axonal sprouting in a defined functional circuit. Cerebellar climbing fibres (CFs) are the axon terminals of inferior olivary neurons in the medulla, innervating Purkinje cells (PCs) in the cerebellar cortex[13]. After partial lesion of inferior olivary neurons, surviving CFs in the cerebellar cortex sprout new collaterals that innervate nearby denervated PCs[11]. Since this innervation process occurs within ∼200 μm of the pial surface in mice, it can be imaged with two-photon in vivo time-lapse microscopy: arguably the most suitable method for studying spatiotemporal aspects of dynamic cellular processes[14,15]. Furthermore, the cerebellar cortex consists of sagittally oriented bands of functional zones. PCs in each zone receive excitatory synaptic inputs from a distinct subset of CFs and send inhibitory outputs to a distinct subregion of the deep cerebellar nuclei[16]. Because of this unique topography of input/output projections, each zone is responsible for a different set of sensorimotor operations[17]. Notably, these sagittal functional zones can be visualized by the expression pattern of zebrin II, which makes this system ideal for an imaging study. A number of molecular markers exhibit sagittally oriented, striped expression patterns with alternating stripes of high and low/no expression[16]. The relationship between these molecularly defined stripes and functional zones is unclear for most markers. However, a recent study shows that zebrin II-positive and -negative stripes receive inputs from functionally distinct group of CFs in mice[18], indicating that the expression pattern of zebrin II represents functional zones in mice.

To study CF collateral sprouting and its relation to the cerebellar functional zones, we used double-transgenic mice in which CFs and zebrin II are labelled with enhanced green fluorescent protein (EGFP) and tdTomato (red fluorophore), respectively. CFs were chemically lesioned by injecting 3-acethypyridine (3-AP) into the inferior olive. Subsequent sprouting of new collaterals and their innervation of PCs was repeatedly imaged over a period of several months using single- and multicolour two-photon in vivo microscopy. We show that synaptic innervation by new collaterals peaks around 4 weeks after the lesion, although the collaterals continue to grow over several months. Importantly, these new collaterals expand their territory beyond the boundary of zebrin II stripes, innervating functionally distinct PCs in neighbouring functional zones. Furthermore, new CFs emerging from these collaterals do not grow uniformly: CFs near the collateral origin gain significantly more synaptic area on their target PCs than distant CFs. These results suggest that although functional boundary does not limit collateral expansion, surviving CFs continue to have their greatest synaptic influence on PCs near their original targets because of the spatial gradient of collateral innervation.

## Results

**Spatiotemporal pattern of collateral sprouting**. We first determined the spatiotemporal pattern of CF collateral sprouting in adult mice. To visualize CFs, we used a transgenic mouse line in which EGFP is expressed under the neurofilament light chain promoter (Nefl-EGFP tg mice). As shown in our previous study[19], ∼80% of CFs in the cerebellar vermis were labelled with EGFP in Nefl-EGFP tg mice (Fig. 1a). In the mature cerebellum, a single CF normally innervates only one PC and closely follows its dendritic arbor attaining an expansive appearance in the parasagittal plane (Fig. 1b, left). In the transverse plane, because of the planarity of the PC dendritic arbor, the CFs have a restricted spread giving them a ladder-like appearance (Fig. 1b, right) with a thick main stalk and thinner rungs extending from the main stalk (Fig. 1c). This ladder-like structure represents the appearance of CFs in our in vivo time-lapse images. We refer to this ladder-like structure as a CF ladder hereafter.

For selective lesioning of CFs, systemic injection of neurotoxin, 3-AP, is commonly used in rats as an experimental model of ataxia[20,21]. However, systemic injection of 3-AP is difficult to apply in mice as the 3-AP dosage necessary to produce CF lesioning is also highly toxic. We therefore injected 3-AP directly into the inferior olive to induce a partial lesion of CFs (Fig. 1d and Supplementary Fig. 1). Similar to a previous report[11], we observed surviving CFs sprouting collaterals within 1 week after the lesion (Fig. 1d). New CF ladders that emerged from sprouted collaterals formed vesicular glutamate transporter 2 (VGLUT2, a well-established marker for CF terminals)-positive varicosities, suggesting that they made functional synapses on dendrites of nearby denervated PCs (Fig. 1e). This is consistent with a previous finding that new CF ladders form synapses with nearby denervated PCs after systemic injection of 3-AP (ref. 22).

Post-lesion CF collateral sprouting in the paravermal region of lobule VI and VII was repeatedly imaged in vivo from 1 week up to 13 weeks after olivary injection of 3-AP ($n = 4$ animals). CFs that emerged from new collaterals had a ladder-like appearance similar to their parent CF (Fig. 2a and Supplementary Fig. 2), indicating that they were innervating dendrites of nearby denervated PCs. Consistent with previous reports[11,23], new CF collaterals always sprouted laterally, expanding the innervation territory of the parent CF only in the mediolateral extent. Since functional zones are oriented sagittally, laterally growing CF collaterals may cross the boundary between functional zones. To quantify the temporal pattern of this mediolateral collateral sprouting, we categorized the newly added CF ladders as outside or inside based on their location of addition when compared with the previous imaging time point. Outside addition refers to the expansion of the mediolateral extent of the parent CF territory, whereas inside addition refers to further innervation of PCs within territory that the CF has already covered (Fig. 2a and

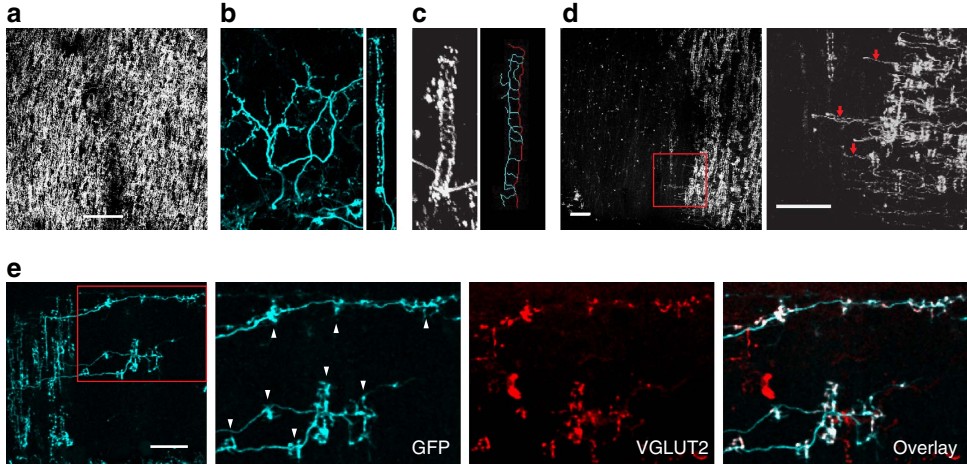

**Figure 1 | CF morphology in the normal cerebellum and CF collateral sprouting induced by 3-AP injection into the inferior olive.** (**a**) Image of normal CFs *in vivo* from a Nefl-EGFP tg mouse (no 3-AP injection) as viewed from a window placed over lobule VII of cerebellar cortex using two-photon microscopy. All *in vivo* images (**a,c,d**) are maximum projections showing top-down views of CFs in the molecular layer. Scale bar, 100 μm. (**b**) Image of normal CFs in a fixed cerebellar slice from a Nefl-GFP tg mouse. View of CFs in the molecular layer in para-sagittal plane (left) and view of another CF in transverse plane (right). (**c**) Image of a CF ladder *in vivo* (left) and its trace (right) in which red line represents the main stalk and the blue lines emerging from the main stalk represent the rungs of the CF ladder. (**d**) A representative image of surviving CFs *in vivo* from a Nefl-EGFP tg mice 1 week after 3-AP injection as viewed from a window placed over lobule VII of cerebellar cortex using two-photon microscopy. (Left) Red square shows site of collateral sprouting magnified in the images on the right. (Right) Red arrows mark collaterals in the magnified view. As shown in this example, collateral sprouting was observed 1 week after 3-AP injection in all animals ($n = 12$). Scale bar, 50 μm. (**e**) Immunolabelling of synaptic sites with VGLUT2 (red) in CFs (anti-GFP, cyan) 2 weeks after 3-AP injection ($n = 2$ animals). The molecular layer of lobule VIII in a fixed coronal section is shown. Red square shows site of collateral sprouting magnified in the images on the right. Note that new CF ladders (white arrowheads) are all VGLUT2-positive regardless of their ladder length and distance from their origin. Scale bar, 50 μm.

Supplementary Fig. 2). Individual, surviving CFs showed variable patterns of inside/outside addition, but they showed a clear temporal profile as a population. Both outside and inside additions continued over several months (Fig. 2b). The rate of outside addition was relatively constant (Fig. 2b, magenta), suggesting that the boundary of functional zones does not limit the mediolateral expansion of the parent CF territory. Inside additions, on the other hand, were more time-dependent since significantly more CF ladders were added inside the mediolateral extents of parent CFs at 4 weeks after 3-AP injection than at any other time points (Fig. 2b, green). Amongst all the 101 ladders analysed, nearly half the ladders (49) were added in the first 4 weeks. Out of these 49 ladders, we could reliably measure the distance between the new ladders and the parent CF for 30 ladders. Out of these 30 ladders, nearly all (27) were within 45 μm of the parent CF. Since the diameter of a mature PC soma is about 15–20 μm, 45 μm is equivalent to the length of only a few PC somata contacted side by side. These results indicate that although the parent CF continues to expand its mediolateral territory over an extended period of time, the majority of new innervations resulting in this mediolateral expansion occurs in the first month and is near the parent CF.

In addition to the mediolateral expansion of parent CF territory, the newly added CF ladders grow sagittally over time, making more synaptic contacts on their postsynaptic PC dendrites. While ladder addition would lead to the parent CF innervating more targets, sagittal ladder growth would allow the parent CF to strengthen its influence on those targets. Therefore, we quantified the rate of ladder growth. For each newly sprouted CF ladder, we measured the length of its main stalk and its total length (main stalk + each rung) (Fig. 3a). The length of the main stalk relates to the ability of new CFs to extend distally along PC dendrites from the initial innervation point, while the total length relates to the overall CF synaptic area on PC dendrites. The change in the length of the main stalk and total ladder over time

showed an almost identical time course as a population. Although each CF ladder appeared at various time points after the lesion, ladder growth was independent of the time after the lesion and was mostly completed within 3 weeks after birth of the ladder (that is, the time point we first observed the ladder in the imaging time series, Fig. 3b). These data suggest that a discrete time window exists in which post-lesion CF collateral sprouting occurs most extensively. This window must occur from 4 to 7 weeks after the lesion, because new ladder addition peaked at 4 weeks after the lesion (Fig. 2b, inset), and these new ladders grew mostly in the next 3 weeks.

**Collateral sprouting in distinct functional zones.** Post-lesion collateral sprouting yields new synaptic connections that are not present under normal circumstances. If postsynaptic targets of new collaterals are functionally relevant to the original targets of parent axons, collateral sprouting might contribute to functional recovery after lesion. On the other hand, if new collaterals randomly innervate any available postsynaptic target without regard to its functional relevance, this is likely to be detrimental to brain function. Therefore, examining the target selectivity of newly sprouted collaterals in relation to functional circuits is crucial for understanding potential consequences of post-lesion axonal remodelling and designing interventional strategies for future therapies. To study post-lesion CF collateral sprouting in relation to the functional architecture, as defined by zebrin II expression, we crossed the Nefl-EGFP tg mice with another line of transgenic mice in which tdTomato is expressed under the aldolase C promoter (Aldoc-tdTomato tg mice; *aldolase C* is the gene that encodes zebrin II)[18,24]. As shown in Fig. 4a, tdTomato-positive and -negative zones are clearly visible, and two-photon multicolour imaging allowed simultaneous visualization of CFs and tdTomato-labelled functional zones in the double-transgenic mice. Boundaries between tdTomato-positive and -negative zones

precisely match with the boundaries identified by zebrin II-expressing (positive) and -non-expressing (negative) zones in Aldoc-tdTomato tg mice[18]. This was also the case in our double-transgenic mice (Supplementary Fig. 3). Consistent with a previous study, which shows that Bergmann glia express zebrin II (ref. 25), we observed tdTomato signal in Bergmann glia (Supplementary Fig. 3). However, this did not interfere with unambiguously distinguishing zones of tdTomato-/zebrin II-positive and -negative PCs (functional zones) in vivo (Fig. 4b).

CF lesion and long-term in vivo time-lapse microscopy were performed as described above except that the imaging was performed in the paravermal region of lobule VIII instead of lobule VI/VII ($n = 6$ animals). We imaged lobule VIII for these

experiments because the zebrin II zones in lobule VI/VII are unclear, whereas the zones were clearly visible in lobule VIII under our cranial window. Before 3-AP injection, EGFP-positive CFs were found in both the zebrin II-positive and -negative zones (Fig. 4a, right). The 3-AP injection caused CF degeneration in both zones but primarily in zebrin II-negative zones. Although we did not intend to perform zone-specific CF lesioning, this is most likely due to our 3-AP injection site since specific subnuclei within the inferior olive project CFs to PCs within specific functional zones in the cerebellar cortex[16]. Importantly, the boundary between zebrin II-positive and -negative PCs was robustly maintained throughout the imaging period at the single-cell level (Fig. 4b). This indicates that although the expression pattern of zebrin II is closely related to the CF projection pattern, neither CF degeneration nor subsequent sprouting affects zebrin II expression in PCs, at least in the mature cerebellum. Therefore, zebrin II-positive and -negative zones in our time-lapse images represent the cerebellar functional structures established before the lesioning.

The temporal pattern of collateral sprouting and ladder addition in the double-transgenic mice was similar to that in the single-transgenic mice (Fig. 5a,b, $n = 3$ animals, only animals that had sufficient time points were quantitatively analysed). Inside ladder addition peaked at 4 weeks after the lesion, and most inside and outside additions were made within the first 4 weeks (Fig. 5b, number of ladders added in the first 4 weeks/total ladders analysed: single transgenic = 49/101, double transgenic = 40/96), suggesting that this temporal profile is preserved across the different lobules. Consistent with our observation from the single-transgenic data, the outside ladder addition did not stop even at 13 weeks post lesion in the double-transgenic mice. More importantly, these newly sprouted ladders added to the outside of the mediolateral extent of the parent CF allowed the parent CF to expand across the boundary of zebrin II stripes, from its original native zone, which is the zebrin II-positive or -negative zone it exists in, into the adjacent non-native zone (Fig. 5c,d). In Fig. 5c, two surviving CFs extend collaterals into adjacent non-native zone, but in Fig. 5d, only one of the two surviving CFs (upper right) poised at the zonal boundary extends a collateral into the non-native zone. In all six double-transgenic animals imaged (including three animals that were excluded from quantitative analysis), some surviving CFs ignored the zonal boundary. Although it is unclear why only a certain subset of surviving CFs extends into a non-native zone,

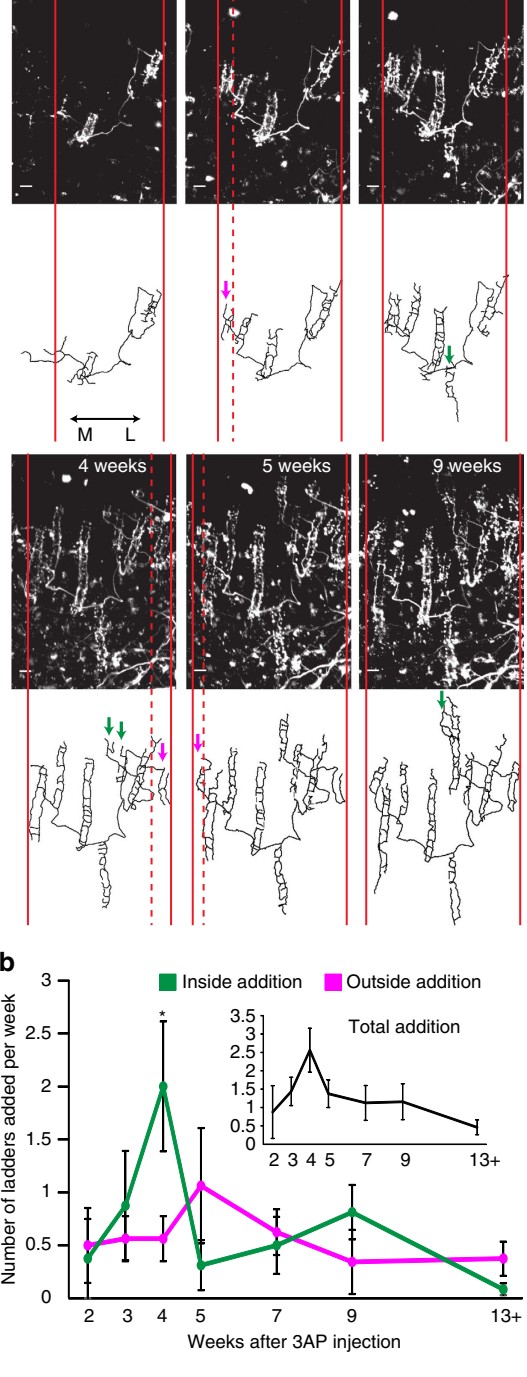

**Figure 2 | Pattern of post-lesion CF collateral sprouting *in vivo*. (a)** A representative example of *in vivo* time-lapse images of the same surviving CF and its traces for the time points mentioned at the top right of the CF images. Maximum projections (top-down view) of the CF in the molecular layer are shown. Solid red lines indicate the mediolateral extent of the CF at each time point while the dashed red line indicates the mediolateral boundary from the previous time point that expanded in the current time point. Magenta arrows indicate ladders categorized as outside additions while green arrows indicate ladders categorized as inside additions. A total of nine surviving CFs were imaged from the four animals and traced completely as shown in this example. Additional examples are shown in Supplementary Fig. 2. Scale bar, 10 μm. **(b)** Average number of ladders added ( ± s.e.m.) at each time point ($n = 4$ animals). We observed 101 new ladders emerging from the 9 surviving, parent CFs out of which 48 were categorized as outside and 53 as inside. The pattern of inside addition (that is, number of ladders added to the inside) were significantly different over time (one-way repeated measures analysis of variance with Tukey *post hoc* analysis: $F(6,18) = 3.604$; $P = 0.01$; $*P < 0.05$ time point 4 compared with time points 2, 5 and 13 + ). Inset shows total ladders added ( ± s.e.m.) at each time point.

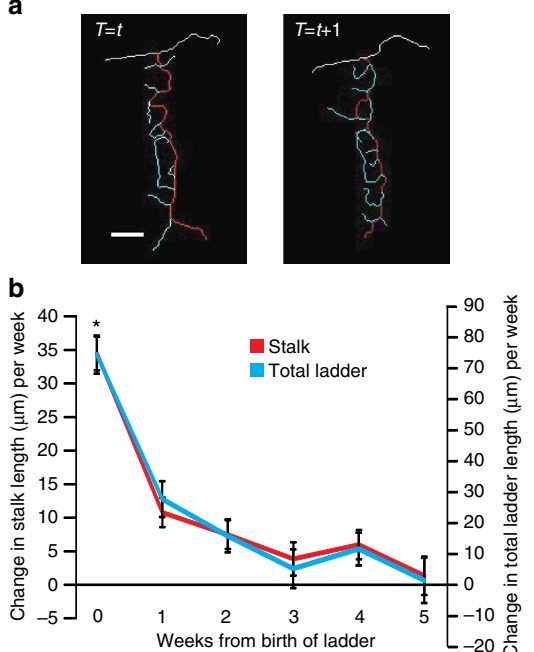

**Figure 3 | Pattern of post-lesion CF ladder growth *in vivo*.** (**a**) Traces from a CF ladder at two consecutive time points (*t* and *t* + 1) are shown to illustrate how change in stalk and total ladder length was measured throughout the paper. The white trace is the main collateral the CF ladder emerges from. The red trace is the main stalk of the CF ladder and the blue traces are the rungs that emerge from the main stalk. Stalk length is the measurement of the red trace while total ladder length is the sum of measurements of the red trace and all the blue traces. Scale bar, 10 μm. (**b**) Average change in stalk length (± s.e.m., red) and total ladder length (± s.e.m., blue) per week at each time point where each time point indicates number of weeks from birth of the ladder (*n* = 57 ladders). Average change in stalk length was significantly different over time (one-way analysis of variance with Tukey *post hoc* analysis: *F*(5, 175) = 26; *P* < 0.0001; *P* < 0.001 compared with time points 1, 2, 3, 4 and 5). Average change in total ladder length was significantly different over time (One-way ANOVA with Tukey *Post-hoc* analysis: F(5,201) = 20.74, *P* < 0.0001, *P* < 0.001 compared to time point 1,2,3,4,5).

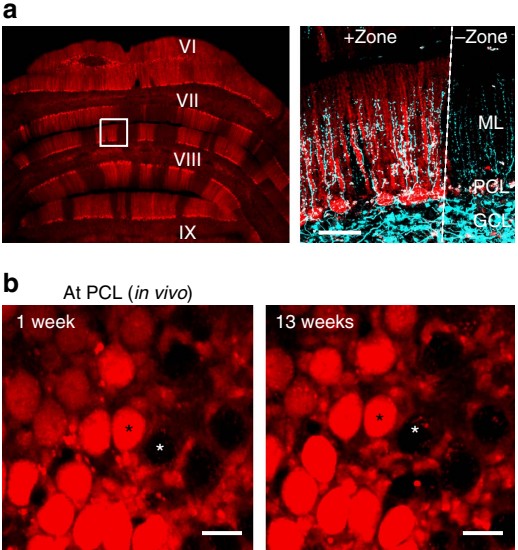

**Figure 4 | Multicolour imaging of zebrin II zones and CFs.** (**a**) Coronal section of cerebellar cortex from Aldoc-tdTomato tg crossed with Nefl-EGFP tg mouse. (**a**, left) Wide-field image of an unfixed, freshly prepared coronal section showing the sagittally oriented bands of zebrin II-expressing (+ zone, red) and non-expressing (-zone) zones. (**a**, right) Magnified image of the area enclosed in the white square from the image on the left. This image was taken using a two-photon microscope to show that EGFP expressing CFs (cyan) can be visualized in both zebrin + zones (red) and –zones. White dashed line indicates the zonal boundary. GCL, granule cell layer; ML, molecular layer; PCL, Purkinje cell layer. Scale bar, 50 μm. (**b**) Representative images taken *in vivo* at the level of the PCL in lobule VIII at 1 and 13 weeks after 3-AP injection show the same PCs as zebrin II-expressing and non-expressing. Zebrin II expression in PCs was stable in all double-transgenic animals that were treated with 3-AP and imaged longer than 4 weeks (*n* = 3 animals). Black and white asterisks indicate examples of zebrin II expressing and non-expressing PCs, respectively. Scale bar, 20 μm.

these data show that the zonal boundary defined by zebrin II expression does not prevent new collaterals from extending into a non-native zone and giving rise to CF ladders in that non-native zone. Furthermore, CF ladders in native and non-native zones both expressed VGLUT2, suggesting that newly formed CF ladders were functional presynaptic terminals regardless of the zones they appeared in (Supplementary Fig. 4).

Robust axonal degeneration, as seen in our experimental set-up, might create a permissive environment that enables axonal collaterals to disregard pre-existing functional zones; though, they might retain a preference for their functionally relevant postsynaptic target. To test this possibility, we compared the growth rate of CF ladders emerging in their native zone with others emerging in non-native zones. We found that the average stalk length and the total length of CF ladders were similar between native and non-native zones when the ladders first appeared in the imaging time series (Fig. 5e,f). Subsequent growth of the ladders was also similar between the zones and, as observed in the single-transgenic mice, the ladder growth was mostly completed within 3–4 weeks after birth of the ladders (Fig. 5e,f). These results indicate that sagittally oriented functional zones in the cerebellar cortex do not affect CF collateral sprouting,

suggesting that surviving olivary neurons are connected to both functionally relevant and irrelevant PCs after the lesion.

**Spatial gradient of synaptic innervation by collaterals.** Since functional boundary did not limit territorial expansion (ladder addition) or strengthening of innervation (sagittal ladder growth) of CF collaterals, we sought to determine if collateral sprouting is affected simply by the distance from the collateral origin. To quantify how the distance from the origin affects collateral sprouting, we measured CF ladder growth in relation to the distance between the ladders and the zonal boundary. Ideally, the distance between the ladders and the collateral origin should be measured, but the origin was difficult to identify in some cases (for example, Figs 1d and 6a). However, as shown in Fig. 6a, the boundary between surviving and degenerated CFs was often very close to the zebrin II zonal boundary in the double-transgenic mice, hence collateral sprouting mostly starts near the boundary in our experimental condition. We therefore used the zonal boundary as the starting point of the measurement. We found that when CF ladders first appeared average stalk length and the total ladder length were negatively correlated with the distance from the boundary, suggesting that ladder growth is slower for distant CFs (Fig. 6b–d). This negative correlation was maintained even after the ladders were fully grown (> 3–4 weeks after birth of the ladder, Fig. 6b–d). These results suggest that although sprouted CF collaterals can innervate distant PCs beyond the

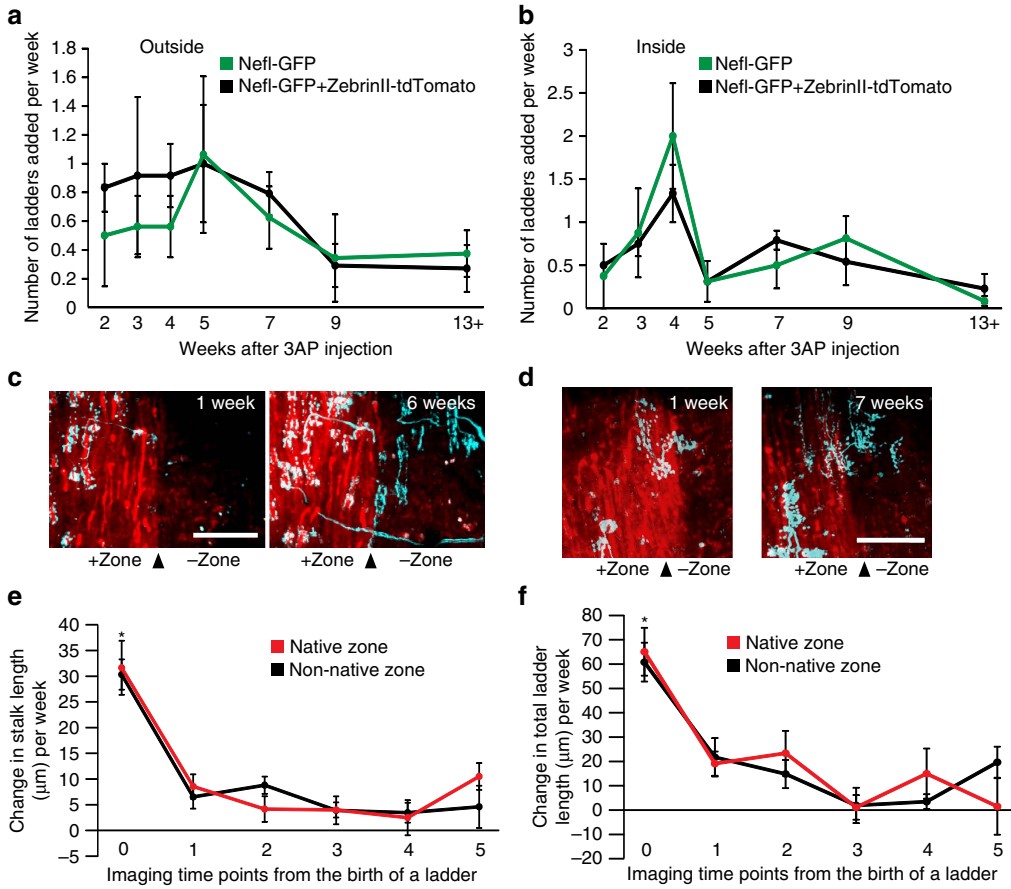

**Figure 5 | Spatiotemporal pattern of CF collateral sprouting in the double-transgenic mice.** (**a**) Average number of ladders categorized as outside added ( ± s.e.m.) at each time point ($n = 8$ surviving CFs from 3 double-transgenic mice, black). The data from single transgenic mice (green, taken from Fig. 2b) are overlaid for comparison. No interaction or genotype or time point effect was found (two-way repeated measures analysis of variance (ANOVA): $F_{(6,30)} = 1.197; P = 0.4$). (**b**) Average number of ladders categorized as inside added ( ± s.e.m.) at each time point ($n = 8$ surviving CFs from 3 double-transgenic mice, black). The data from single-transgenic mice (green, taken from Fig. 2b) are overlaid for comparison. No interaction or genotype effect was found, however time points did have a significant effect (two-way repeated measures ANOVA: $F_{(6,30)} = 3.125; P = 0.01$). (**c,d**) Representative images showing CF collaterals crossing the zonal boundary in lobule VIII. *In vivo* two-photon image of the same area of cerebellar cortex taken 5/6 weeks apart. Maximum projections (top-down view) of the CFs and the zonal boundary in the molecular layer are shown. Scale bar, 50 µm. (**e**) Average change in stalk length ( ± s.e.m.) of ladders growing in native (red, $n = 49$ ladders) and non-native (black, $n = 32$ ladders) zones at each time point, where each time point indicates number of weeks from birth of the ladder. The change in stalk length was significantly different over time for ladders growing in their native or non-native zone (one-way ANOVA with Tukey *post hoc* analysis: native zone; $F_{(5,116)} = 7.19, P < 0.0001$, *$P < 0.001$ compared with time points 1, 2, 3, 4 and 5; and non-native zone; $F_{(5,99)} = 17.5, P < 0.0001$, *$P < 0.001$ compared with time points 1, 2, 3, 4 and 5). (**f**) Average change in total ladder length ( ± s.e.m.) of ladders growing in native (red, $n = 49$ ladders) and non-native (black, $n = 32$ ladders) zones at each time point, where each time point indicates number of weeks from birth of the ladder. The change in total ladder length was significantly different over time for ladders growing in their native or non-native zone (one-way ANOVA with Tukey *post hoc* analysis: native zone; $F_{(5,116)} = 7.667, P < 0.0001$, *$P < 0.001$ compared with time points 1, 2 and 3; and non-native zone; $F_{(5,99)} = 8.137, P < 0.0001$, *$P < 0.001$ compared with time points 1, 2, 3 and 4).

functional boundary, the surviving, parent CFs have more synaptic influence on nearby PCs than distant PCs.

## Discussion

In this study, we used long-term two-photon *in vivo* time-lapse microscopy to address two crucial aspects of post-lesion axonal remodelling in the mature mammalian brain; its spatiotemporal pattern and its target selectivity in reference to functional circuits. The cerebellar CFs are a particularly suitable model for this purpose because they have several unique advantages over other axons in the brain. First, lesion-induced CF sprouting is an established model of post-lesion axonal remodelling and can be imaged *in vivo*. Second, synaptic innervation by CFs can be visually identified because of the

characteristic morphology of the CFs (sagittally oriented ladders) on their postsynaptic PCs. Electron microscopy analysis of synapses between the new CF ladders formed post lesion and their synaptic targets show that these are typical asymmetric synapses with morphology that is characteristic of this synapse in a normal brain[22]. Third, the functional architecture of local circuits can be visualized by the expression pattern of zebrin II, which uniquely allows imaging of axonal collateral sprouting in functional circuits. Our main findings are that surviving CFs continuously innervate new PCs over several months, although most new innervation occurs near the surviving CFs within the first 4 weeks after the lesion; newly sprouted CF collaterals that originate from one functional zone innervate new target PCs in adjacent functional zones; and the growth of new CFs depends

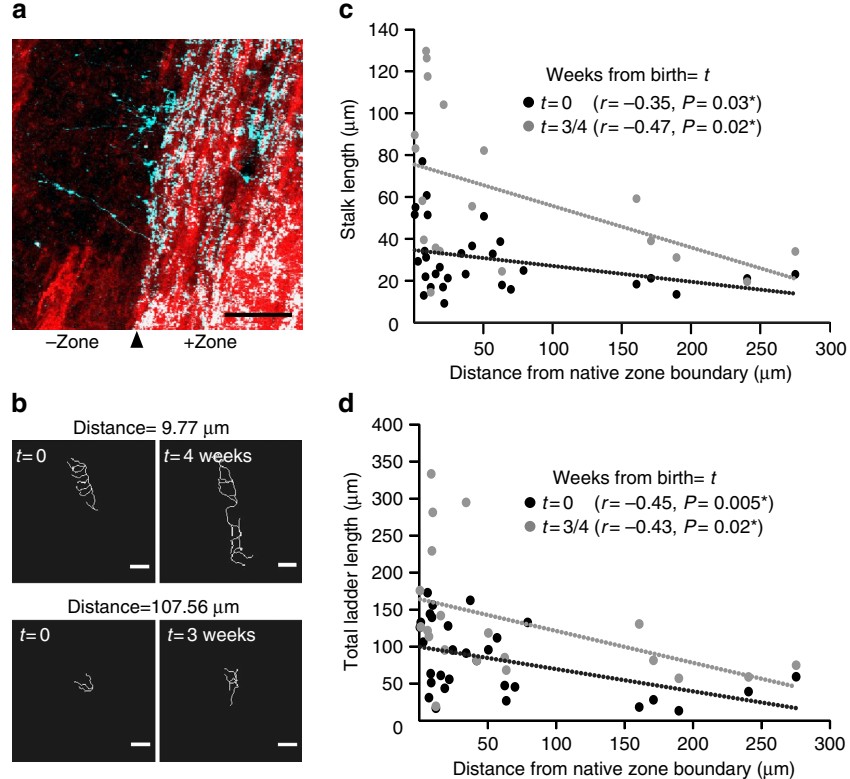

**Figure 6 | Correlation between CF ladder growth and distance from zonal boundary.** (**a**) Representative *in vivo* image showing axonal degeneration in lobule VIII occurs at the boundary of positive and negative zebrin II zone. A maximum projection (top-down view) of the CFs and the zonal boundary in the molecular layer is shown. Four out of the six double-transgenic animals showed similar type of CF degeneration (zonal degeneration) after 3-AP injection. Scale bar, 50 μm. (**b**) Traces of CF ladders growing very close (upper traces) or far (lower traces) from zonal boundary at two time points about 3 or 4 weeks apart. Distance mentioned above the images is measured from native zone boundary. Scale bar, 10 μm. (**c**) Correlation between stalk length at birth of the ladder (*t* = 0, black circles, *n* = 30 ladders) or at 3 or 4 weeks from the birth of the ladder (*t* = 3/4, grey circles, *n* = 20 ladders) and distance from native zone boundary. (**d**) Correlation between total ladder length at birth of the ladder (*t* = 0, black circles, *n* = 30 ladders) or at 3 or 4 weeks from the birth of the ladder (*t* = 3/4, grey circles, *n* = 20 ladders) and distance from native zone boundary.

on the distance from the collateral origin to the new CFs, where distant CFs show slower and less growth than adjacent CFs.

In the first report of lesion-induced CF collateral sprouting, Rossi *et al.*[11] performed conventional histochemical analyses using fixed tissues and found that CFs in lesioned cortical sections had larger total surface area when compared with un-lesioned CFs at 1 month after 3-AP injection[11]. The authors mention that due to small sample size they could not statistically analyse this growth, but their data suggested that CF growth continued for several months. Our data are consistent with their results and provide new insights into the spatial pattern of CF collateral sprouting. We show that newly sprouted CF collaterals do not sequentially innervate from adjacent to distant PCs. Rather, CF collaterals may innervate PCs not adjacent to the origin first, and then innervate closer PCs (inside addition) later. It is unclear why the sprouting collaterals initially ignore some PC dendrites that they will eventually innervate. Future studies are required to identify the mechanism that initiates synaptic innervation and the role of the postsynaptic neuron in this process.

Repeated imaging of the same surviving CFs over time allowed us to identify the detailed temporal profile of CF sprouting. We show that new synaptic innervation most frequently occurs in a narrow time window at around 4 weeks after the lesion. In addition, the newly formed CF ladders grow most rapidly within a few weeks after the onset of innervation. Taken together, we can conclude that post-lesion CF collateral sprouting (both ladder addition and ladder growth) most extensively occurs around 4–7 weeks after the lesion. This time course seems significantly slower

than collateral sprouting in the adult macaque primary visual cortex. Yamahachi *et al.*[26] used two-photon *in vivo* time-lapse microscopy to observe retinal lesion-induced axonal sprouting of layer 2/3 pyramidal neurons in primary visual cortex. They found that rapid axonal sprouting occurred even on the day of the lesion, and that the density of newly sprouted axons reached its peak within the first week[26]. Interestingly, the density of the cortical axons declined afterwards due to axonal pruning, which was not the case for CF sprouting in our experiments. Although we found that a small number of new CF ladders later disappeared or significantly decreased in length (Supplementary Fig. 2b, blue arrowhead), they were a minor population; the majority of new CFs persisted throughout the imaging period. These results indicate that the temporal profile of post-lesion axonal sprouting differs depending on axon type, brain region and type of injury.

In many brain regions, functionally related neurons are located close to each other resulting in distinct clusters that give rise to a unique functional architecture. It is fundamental to understand how lesion-induced collateral sprouting proceeds in relation to such a functional structure. Visualizing brain functional structure usually requires measuring neuronal activity in response to a distinct stimulus. However, in our case we took advantage of the expression pattern of zebrin II to visualize cerebellar functional structure. Recent studies show a number of functional differences between zebrin II-positive and -negative zones. PCs in zebrin II-positive zones show impaired metabotropic glutamate receptor-dependent synaptic plasticity due to high expression of

glutamate transporter in those zones[27,28]. CFs projecting to zebrin II-positive zones show higher glutamate release than those projecting to zebrin II-negative zones[29]. Frequency and regularity of simple spikes (one form of PC's firing activity) differ between zebrin II-positive and -negative zones[30]. Complex spikes (the other form of PC's firing activity, synaptically driven by CFs) are highly synchronized within a single zebrin II zone, but not across zones[18]. These studies collectively indicate that PCs in zebrin II-positive and -negative zones have distinct functional properties and receive functionally distinct group of CF inputs. Therefore, although it is possible that a single zebrin II zone could be further divided into multiple smaller functional zones[16], different zebrin II zones represent distinct ensemble of PCs that have at least some, if not all, of the same functional characteristics.

Using our double-transgenic mice, we identified the relationship between the process of CF collateral sprouting and the pre-lesion functional structure. A previous study also tried to examine this relationship using fixed-tissue comparisons. In that study, tissue samples were collected several months after 3-AP injection and surviving CFs were co-labelled with zebrin II. Since many CF ladders were located near the zebrin II boundary and CF collaterals rarely crossed the boundary, the authors reasonably concluded that newly sprouted CF collaterals do not cross the zebrin II boundary[23]. However, the same results could be obtained even if CF collaterals can cross the boundary, if the expression pattern of zebrin II itself changes in accordance with the distribution of the newly formed CFs or CF collaterals that innervate into adjacent zebrin II zone (non-native zone) are later pruned. *In vivo* time-lapse microscopy provides the best experimental approach to test these possibilities. Surprisingly, our data showed that newly sprouted CF collaterals did cross the zebrin II boundary and innervated PCs in non-native zones. The growth of new CF ladders in native and non-native zones were almost identical. Furthermore, the expression pattern of zebrin II was robustly maintained at the level of the single-cell and CF ladders in non-native zones were not pruned during the imaging period of several months. These seemingly inconsistent results between our study and the previous study might be due to the difference in animal species (mouse versus rat) or method of 3-AP administration (local versus systemic injection). In particular, the systemic injections of 3-AP that were used in the previous study killed 90–99% of CFs[11,23], whereas the olivary injection of 3-AP used in our study killed only a small population of CFs. Therefore, the overall integrity of the olivocerebellar circuits is more preserved in our lesion model. Nevertheless, at least in mice, newly sprouted CF collaterals ignore the zonal boundary and innervate new target PCs that are functionally unrelated to the original targets. Such aberrant synaptic connections are likely to be unfavourable for brain function. However, it should be noted that a surviving CF has more synaptic influence on PCs near their original target than distant targets. This suggests that a majority of targets for a surviving CF might mostly be located nearby, within the same functional zone unless the surviving CF is located near the zonal boundary.

Another important question is whether the overall effect of post-lesion CF collateral sprouting is beneficial to animals. The 3-AP injection causes ataxia, but our mice showed gradual recovery afterwards without any treatment. However, it is difficult to correlate the behavioural change with CF collateral sprouting in our experiment because we can image only a small portion of the cerebellar cortex and currently do not have any tool to experimentally manipulate the sprouting process. An important insight can be obtained from another form of post-lesion CF collateral sprouting. When olivocerebellar axons are transected at one side of the inferior cerebellar peduncle, CFs in the ipsilateral side of the cerebellum degenerate, but surviving olivocerebellar axons from the other side sprout new collaterals in the white matter, cross the midline and give rise to new CFs on the denervated side[31]. This transcommisural CF re-innervation spontaneously occurs if the lesion is performed during the first postnatal week and can be induced up to the third postnatal week by injecting trophic factors such as brain-derived neurotrophic factor and insulin-like growth factor into the denervated side of the cerebellum[32,33]. Since collateral sprouting in the present study was observed in adult animals and is methodologically different from transcommisural CF re-innervation, comparing the spatiotemporal patterns of those two different forms of post-lesion CF collateral sprouting may not be informative. Nevertheless, two aspects of transcommisural CF re-innervation provide insight into the functional significance of post-lesion CF collateral sprouting. First, when transcommisural CF re-innervation occurs, the location of the new CFs and the surviving CFs is symmetrical across the midline, indicating that new CFs innervate zones functionally related to their zone of origin[34–36]. Second, transcommisural CF re-innervation restores motor deficits[8,37]. Therefore, post-lesion CF collateral sprouting is suggested to be beneficial as long as newly formed CFs innervate targets functionally relevant to the original surviving CFs. In our study, since the majority of surviving CFs innervate targets nearby, most likely within the same functional zone, some amount of beneficial functional recovery might be possible even in adult animals.

In summary, we speculate that newly sprouted axon collaterals do not selectively innervate functionally relevant postsynaptic neurons in the mature brain. However, a surviving axon has more synaptic inputs on neurons near the axon's original postsynaptic target(s). Because of this spatial gradient, functional circuits might be loosely maintained in a brain area in which functionally relevant neurons are clustered together. In such a case, post-lesion collateral sprouting would be beneficial for restoring a portion of lost brain function.

## Methods

**Animals.** All procedures were approved by the Institutional Animal Care and Use Committee of the University of Texas at Austin. The Nefl-EGFP tg mice were purchased from the Mutant Mouse Resource Research Centers (stock number: 015882-UCD). The Aldoc-tdTomato tg mice were produced as described previously in Tsutsumi et al.[18]. Double-transgenic mice were obtained by crossing Nefl-EGFP tg mice with Aldoc-tdTomato tg mice.

**3-AP injection and cranial window surgery.** Adult (3–6 months old, both sexes) single and double-transgenic mice were anaesthetized with an intraperitoneal injection of ketamine/xylazine (100/10 mg kg$^{-1}$). The stereotaxic injection to the inferior olive was performed as described previously with a slight modification to the stereotaxic coordinates[14]. To lesion CFs that project to lobule VI and VII, a single 3-AP injection was made at the midline, at the midpoint between the caudal edge of the cerebellar cortex and the C1 cervical vertebra, at a depth of 1.7–1.8 mm. The injection pipette was angled 52° from vertical and 7° from the midline towards the left inferior olive. To lesion CFs that project to lobule VIII, two 3-AP injections were made at 0.3 and 0.6 mm left of the midline (a single injection at each site), at a depth of 1.8 mm (0.3 mm left) and 1.9 mm (0.6 mm left). The injection pipettes were angled 50° (0.3 mm left) and 46° (0.6 mm left) from vertical. A volume of 0.1–0.2 μl of 3-AP (A21207, 1.1 g ml$^{-1}$, Sigma Aldrich, St Louis, MO) was delivered over 5–10 min per each injection site with a Nanoject II automated nanolitre injector (Drummond Scientific Company, Broomall, PA). The pipette was then left in place for 5 min before being withdrawn. Immediately following the 3-AP injection, a small rectangular cranial window ($\sim 2 \times 1.5$ mm$^2$) was made over the right paravermal region of the cerebellar cortex as described in our previous publications[19,37]. Briefly, the muscles and fascia overlaying the skull were removed and the skull surface was cleaned. A thin layer of surgical cyanoacrylate was applied to the dried skull surface, and then a small metal plate was attached near lambda with dental cement. The metal plate was used to securely hold the animal's head during the surgery and subsequent imaging sessions. A rectangular craniotomy was made using a dental drill, and a coverslip (slightly smaller than the size of the craniotomy) was placed directly on the intact dura. The coverslip was first secured by surgical cyanoacrylate and then by dental cement, which was applied around the

coverslip and on the exposed skull surface. The animals were then allowed to recover from the anaesthesia and returned to their home cage.

**Immunohistochemistry and histology.** For immunohistochemical analysis, animals were anaesthetized with an intraperitoneal injection of ketamine/xylazine (100/10 mg kg$^{-1}$) followed by transcardial perfusion with 4% paraformaldehyde in 0.1 M phosphate-buffered saline (PBS). Post perfusion, the brain was extracted and further fixed overnight by immersion fixation in 4% paraformaldehyde in 0.1 M PBS at 4 °C. After washing (three times with 0.1 M PBS at room temperature), 40 µm-thick coronal sections of the cerebellum were prepared using a vibrating microtome (HM650V, Thermo Fisher Scientific, Waltham, MA) for immunohistochemical analysis. Slices were first permeabilized using PBS with 0.3% Triton-X100 (PBST) for 20 min at room temperature, followed by blocking in 5% normal goat serum (121517, Jackson ImmunoResearch, West Grove, PA) in PBST for 1 h at room temperature. Primary antibodies were prepared in blocking reagent and incubation was done overnight at 4 °C. After washing (three times with PBST at room temperature), secondary antibody incubation was performed in blocking reagent for 1 h at room temperature. After washing, slices were mounted (Permafluor, Thermo Fisher Scientific, Waltham, MA), coverslipped and imaged using laser-scanning confocal microscope (FV-1,000, Olympus, Tokyo, Japan). To avoid bleed through between fluorophores, images were acquired sequentially using 405, 488 and 543 nm lasers. Z-stack of images were acquired at 1–2 µm step size. Primary antibodies used were rabbit anti-aldolase C (1 µg µl$^{-1}$), rabbit anti-3-phosphoglycerate dehydrogenase (3PGDH, 1 µg µl$^{-1}$), mouse anti-VGLUT2 (1:400, 135421, Synaptic Systems, Goettingen, Germany) and rabbit anti-GFP (1:500, A11122, Thermo Fisher Scientific, Waltham, MA). Anti-aldolase C and anti-3PGDH antibodies were kindly provided by Masahiko Watanabe at Hokkaido University, Japan[38,39]. Secondary antibodies used were Alexa Flour 405-labelled goat anti-rabbit antibody (A31556), Alexa Fluor 488-labelled goat anti-rabbit antibody (A11034) and Alexa Fluor 405-labelled goat anti-mouse antibody (A31553, Thermo Fisher Scientific, Waltham, MA) at 1:1,000 concentration. For fluorescent Nissl staining, 40 µm-thick coronal sections of medulla were stained with NeuroTrace 435/455 (N21479, Thermo Fisher Scientific, Waltham, MA) according to the protocol provided by the manufacturer.

**In vivo imaging.** Long-term two-photon in vivo time-lapse microscopy was performed as described in our previous publications[19,40]. Briefly, 1 week following the 3-AP injection, mice were lightly anaesthetized with 1–1.5% isoflurane and securely placed on a custom-made microscope stage. The stage was then fixed on an x–y translator under a two-photon laser-scanning microscope (FV1000MPE, Olympus, Tokyo, Japan) equipped with a × 25 water immersion objective lens (Olympus XLPlan N, 1.05 numerical aperture) and two external gallium arsenide photodetectors (GaAsPs, Hamamatsu, Japan). For two-photon excitation of EGFP and simultaneous excitation of EGFP/tdTomato, 920 nm of pulsed infrared laser was provided by Mai Tai HP DeepSee mode-locked Ti:sapphire laser (Spectra-Physics, Santa Clara, CA). The emitted green and red fluorescent signals were separated by a dichroic mirror (a longpass filter at 570 nm) and then filtered by emission filters (bandpass filters 495–540 and 570–620 nm for green and red emissions, respectively) before being detected simultaneously by the GaAsPs. The z-stack images (spaced 1 µm apart) of CFs and CFs/zebrin II in the cerebellar molecular layer were taken with a resolution of 0.34–0.44 µm pixel$^{-1}$ in most images. Brain surface vasculature was also imaged in wide-field fluorescence mode, to enable us to locate the same CFs in subsequent imaging sessions. In addition, mossy fibres in the granule cell layer (they also express EGFP as shown in Fig. 4a right) and PC somata (in the case of double-transgenic animals) were always included in the z-stack images of CFs. The spatial pattern of mossy fibres and; tdTomato expressing and non-expressing PC somata were extremely stable and unique to each field of view. The unique pattern of these neuronal elements, immediately below the CFs, ensured that the same CFs were imaged across every session. After image acquisition, the animals were allowed to recover from the anaesthesia and returned to the home cage.

**Image analysis and statistics.** Images were analysed using the simple neurite tracer plugin in Fiji, an ImageJ-based open-source image-processing package (http://fiji.sc/Fiji). Surviving CFs were identified at the first time point (1 week after 3-AP injection) and traced completely at each time point. Tracing was done on a z-stack of images. The main stalk of each ladder was identified using a semi-automatic system built into the plugin used. We located the two ends of the main stalk in the following manner: first, we selected one of the farthest visible points of the ladder in the z-plane and then selected the other end of the stalk as the farthest point of the ladder from the first selected point in the x and y planes. The plugin then automatically found the connecting fibre between these two selected points, which we considered the main stalk of the ladder. The rungs of the ladder were then traced semi-automatically by selecting the point on the rung farthest from the main stalk and then letting the plug-in trace where this rung connects to the main stalk automatically. This semi-automatic approach allowed us to quickly trace fibres as they weave through many z-planes. The plugin also kept track of measurements for the main stalk and the rungs, which we used for all our analyses. Inside and outside ladder additions were categorized as has been explained in the

results. Specific statistical analysis performed and sample sizes are mentioned in the figure legends wherever required. Six single-transgenic animals and six double-transgenic animals were imaged for the experiments. Among them, four single-transgenic animals and three double-transgenic animals were quantitatively analysed. These sample sizes were not pre-determined by any statistical methods but were chosen on the basis of what is normally reported in similar long-term in vivo time-lapse imaging publications[14,41–43]. All animals that showed significant CF damage and denervation under the optical window at the first imaging time point (1 week after injection) were imaged for the rest of the imaging sessions. Out of the six single-transgenic animals imaged, two were excluded because of poor image quality at the later time points (4 weeks onwards). Out of the six double-transgenic animals imaged, two were excluded because they were imaged only for 4 weeks due to bone regeneration under the cranial window. One was excluded because of poor image quality at the later time points (4 weeks onward). No randomization or blinding was necessary since all animals received exactly the same treatment. For all analyses of variance, normality of data was confirmed using Shapiro–Wilk normality test, and equality of variance between groups was confirmed using Bartlett's test. An estimate of variance within each group was calculated and is reported in the relevant figures as error bars (s.e.m.) and in the result section. All time course data are presented as average ± s.e.m.

**Data availability.** The authors declare that the data supporting the findings of this study are either available within the article (and its Supplementary Information file) or available from the corresponding author on request.

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

## Acknowledgements

We thank Masahiko Watanabe (Hokkaido University, Japan) for providing anti-aldolase C and anti-3PGDH antibodies; Naoko Nishiyama for technical assistance and maintenance of mouse colonies; Jeremy Colonna and Elise Shen for assistance during the initial stage of this study; and Alex Fanning and Moushumi Dey for helpful comments on this manuscript. This work was supported by National Institutes of Health Grant NS073919 (H.N.) and Grants-in-Aid for Scientific Research (25000015 to M.K. and 24240048 to K.S.) from Japan Society for the Promotion of Science.

## Author contributions

M.D. and H.N. designed the experiments; M.D., J.M.B. and H.N. performed immuno-histochemistry, surgeries and image acquisition; M.D. and J.M.B. performed data analysis; M.D., J.M.B and H.N. wrote the manuscript; K.S. and M.K. generated Aldoc-tdTomato tg mice and helped with the manuscript.

## Additional information

**Competing financial interests:** The authors declare no competing financial interests.

