## [Peer Review File · Nature Communications]

Reviewers' comments:

Reviewer #1 (Remarks to the Author):

The paper by Dhar et al. explores lesion-induced axonal remodeling 'live', using two-photon microscopy, in the brain in vivo. The authors make an intelligent use of the particular circuitry structural features of the cerebellar cortex to answer a major but unresolved issue in neuroscience: when axon collaterals grow out, do they target functionally relevant neurons or are initial connectivity patterns simply formed at random? The present study clearly demonstrates that the latter alternative is the correct answer.

The study is technically advanced, and have been conducted with great technical skill. Axonal outgrowth is generated by killing off a small number of neurons in the inferior olive in the brain stem, which only contains neurons projecting climbing fibers to the cerebellar cortex. The dead axons leave some Purkinje cells climbing fiber-denervated and axonal outgrowth is triggered in the nearby climbing fiber axons. Using in vivo two-photon microscopy, the outgrowth of specifically labelled climbing fibers are visually monitored for up to 13 weeks. Some climbing fiber lesion studies have previously been conducted, but this is the first time that the progress of the sprouting is followed 'live'. The techniques used here also give a better resolution, and the functional issue of which Purkinje cells a climbing fiber sprout targets was not so well described before.

The manuscript is well written and the data presentation and figures are of good quality. Statistical analyses is not the main approach here, where the results are reported in morphological terms. The title and the abstract reflect well the contents of the paper and the conclusions that can be drawn.

I have only one minor issue, which the authors might want to clarify:

In the heroic experiment of following climbing fiber collateral sprouting for 13 weeks, it is stated that the sprouts of an individual climbing fiber were followed. However, I find little explanation of how the authors could verify that this was really the same climbing fiber. I assume it is based on the window location and some form of congruence in the morphology of the climbing fiber terminal tree/ladder system. Depending on the degree of congruence, it may be highly suggestive, but doesn't prove that it is the same climbing fiber from one session to another. Small brain movements between sessions could in principle result in that another climbing fiber is brought into the center of the window. I think it could be helpful if the authors expand a little bit on the motivation why it can be assumed that one and the same climbing fiber can be analyzed throughout sessions.

Reviewer #2 (Remarks to the Author):

MS ID#: NCOMMS-16-01043

MS TITLE: Spatiotemporal dynamics of lesion-induced axonal remodeling and its relation to

functional architecture of the brain

AUTHORS: Dr. Nishiyama and co-workers

In this manuscript, the authors used in vivo time-lapse microscopy and mouse transgenic reporter lines to re-examine the behavior of climbing fiber collateral re-growth in the adult cerebellum. The technology is excellent and the area of research is interesting.

Although the authors present some intriguing data, I am confused about the interpretations of the results. My feeling is that this confusion arises because the authors would like to make a strong case that their results argue against the previous works from the Hawkes, Rossi, and Sherrard groups. But based on the data that is shown here, I am not so sure the results are quite so different from these earlier reports. But even if they are different as suggested, then a question has to be raised about the very different methodological approaches that were taken to disrupt the climbing fiber system.

Major criticisms:

1) This manuscript is driven by the argument that climbing fiber collateral sprouting after lesioning the olive does not selectively re-innervate the cerebellar zones that were targeted before lesion. However, I am concerned about how the zones are delineated, since the tdTomato expression (a proxy for ZebrinII expression) shows something peculiar. The zones defined by this transgenic are not so sharp. If this is really the case and not just an imaging artifact, then climbing fibers that are normally targeted to these "red" Purkinje cells would also sprout to them - in panel 5C you can see Red labeling in supposedly negative zones. That is, the boundaries between zones are not as sharp as one would expect of zebrinII expression. Therefore, it's not so surprising to me that you would get sprouting to both zones given that the boundaries, as defined here, are not so well defined in the first place.

2) I am concerned that the manipulation is not specific enough to make claims about zone specific ablation and regrowth. The authors mention that after 3-AP injection climbing fibers in both the positive and negative zebrinII zones (adjacent compartments) show some degree of climbing fiber degeneration, but I suppose the degeneration is more predominant in one zone over the neighboring one. Still, why then is it surprising that you would get regrowth into both compartments? Even if a single fiber can be tracked and is found to cross a boundary, I am not sure what this means since the whole compartmental integrity is compromised. That is, even if the authors are correct, it's not so surprising that zones become irrelevant when there is massive destruction of the circuit.

3) Based on the two points above, and the fact that the methodology used in this study is completely different from the other groups (and others used rat), I am not sure such strong comparisons can be made. In fact, the design of the study here seems like this paper would address a totally different set of questions to me. In end, this paper likely tells us more about competitive rewiring of a general territory when innervation to multiple compartments is disrupted rather than a zonal specific story.

4) In the discussion, the authors spend a whole paragraph discussing the BDNF work from Sherrard, almost in support of their data. This is confusing because Dr. Sherrard made the claim for zonal re-growth. But, Nishiyama and colleagues suggest that even though non-native zones are re-innervated, there is still a preference for the native zone. As I mentioned above, is there really any major difference between what they see and what was previously reported? Point number (3) at the end of the first paragraph in the Discussion section basically hints at similar outcomes. Even if the collaterals tend to go to a nearby zone, ultimately they still have a preferential target, even if proximity is the guiding factor. Thus, the take home message is confusing because a solid interpretation cannot be made.

5) Injecting 3-AP directly into the olive seems like a major perturbation. How was the spread of the toxin controlled? I would assume that this method kills more than just the native and non-native adjacent zones that are described. The authors need to provide some detailed histological characterization of what these injections do - for example, cell death in the olive and also within cerebellar cortex, low power images of the injection spot in the olive to show the damage done because of the toxin and also because of physical/mechanical damage from the injection itself. What about adjacent nuclei such as the lateral reticular nuclei? Are they damaged too? Do they re-grow? Some of the profiles in panel 5C look like ectopic mossy fibers in the molecular layer (look at the fiber and terminal just to the left of the 6 in the label "6weeks"). Perhaps some of the abnormal growth that is seen is from mossy fiber collateral regrowth? 200 nanoliter injection is quite large, so a clear and thorough analysis of regional pathology should be provided.

Additional concerns

1) The title of the paper is much too broad. There is no mention of cerebellum, climbing fiber, zones, module, etc. I am also not sure what the authors mean by "axonal remodeling".

2) In the figure panels, the layers of the cerebellar cortex should be labeled. This will give readers that are not in the cerebellar field some context to work with.

3) Along the same lines as above, even though it is mentioned in the methods which lobules are targeted, the lobule numbers should also be labeled on the images.

4) It is incorrect to refer to cerebellar zones as "slabs".

5) The number of animals used for each experiment should be mentioned in the main body of the text.

Reviewer #3 (Remarks to the Author):

Axonal remodeling is a recognized cause and consequence of many neurological disorders. It may contribute to the formation of aberrant connections that elicit pathological network

activity or to (partial) re-innervation of synaptic targets. In this study, Dhar and colleagues study lesion-induced remodeling of cerebellar climbing fibers (CFs) using an elegant in vivo approach that relies on 2-photon time-lapse imaging of genetically labeled CFs following 3-AP injections into the inferior olive. The authors provide a very detailed description of how CF sprouting occurs, both temporally and spatially. This work provides several anatomical insights such as that sprouting CFs innervate relevant and non-relevant Purkinje cells (PCs) and that they do not obey to functional boundaries such as those marked by zebrin II.

This is a carefully conducted study that confirms and expands previous (immuno)histological data on CF sprouting. The technical approach used is elegant and has a lot of potential for the future dissection of mechanisms underlying CF sprouting. Unfortunately, the current study is purely descriptive and goes into excessive detail on how CF sprouting occurs. However, in my opinion no major insights into CF sprouting are presented. Such insight would require combining the presented tools (e.g. Nefl-eGFP mice, 2-photon imaging) with manipulation or visualization of specific cellular processes or molecules relevant to axonal remodeling.

Figure 2: I would suggest the authors to add more examples of CF sprouting. The graph in Figure 2B summarizes a lot of different examples of CF sprouting but it is unclear how similar or different the spatial patterns of CF remodeling can be. Do CFs always generate the same number of ladders?

Throughout the manuscript, the authors state that CF collaterals generate new synaptic connections. I believe this idea is based on the characteristic morphology of the collaterals? However, how do the authors know that synapses are formed and are these different for relevant and non-relevant PCs?

Perhaps the authors can add additional labels to Figures 5A and 5B to emphasize the difference between these two graphs (inside vs outside addition).

Point-by-point response to reviewers for Dhar et al., NCOMMS-16-01043.

We thank the reviewers for the thoughtful and constructive comments that were very helpful for improving our paper. We also thank the editor for the helpful instructions on how to address the reviewers' comments. In this revision, we have fully addressed these valuable comments by adding new experiments, performing additional data analyses, and improving the clarity of the manuscript. The reviewers' comments are written in black, and our responses are written in blue below.

Reviewer #1:

The paper by Dhar et al. explores lesion-induced axonal remodeling 'live', using two-photon microscopy, in the brain in vivo. The authors make an intelligent use of the particular circuitry structural features of the cerebellar cortex to answer a major but unresolved issue in neuroscience: when axon collaterals grow out, do they target functionally relevant neurons or are initial connectivity patterns simply formed at random? The present study clearly demonstrates that the latter alternative is the correct answer.

The study is technically advanced, and have been conducted with great technical skill. Axonal outgrowth is generated by killing off a small number of neurons in the inferior olive in the brain stem, which only contains neurons projecting climbing fibers to the cerebellar cortex. The dead axons leave some Purkinje cells climbing fiber-denervated and axonal outgrowth is triggered in the nearby climbing fiber axons. Using in vivo two-photon microscopy, the outgrowth of specifically labelled climbing fibers are visually monitored for up to 13 weeks. Some climbing fiber lesion studies have previously been conducted, but this is the first time that the progress of the sprouting is followed 'live'. The techniques used here also give a better resolution, and the functional issue of which Purkinje cells a climbing fiber sprout targets was not so well described before.

The manuscript is well written and the data presentation and figures are of good quality. Statistical analyses is not the main approach here, where the results are reported in morphological terms. The title and the abstract reflect well the contents of the paper and the conclusions that can be drawn.

Our response: We are truly thankful for the reviewer's highly positive evaluation of our work. Below is the reviewer's specific comment and our response to it.

1) I have only one minor issue, which the authors might want to clarify: In the heroic experiment of following climbing fiber collateral sprouting for 13 weeks, it is stated that the sprouts of an individual climbing fiber were followed. However, I find little explanation of how the authors could verify that this was really the same climbing fiber. I assume it is based on the window location and some form of congruence in the morphology of the climbing fiber terminal tree/ladder system. Depending on the degree of congruence, it may be highly suggestive, but doesn't prove that it is the same climbing fiber from one session to another. Small brain movements between sessions could in principle result in that another climbing fiber is brought into the center of the window. I think it could be helpful if the authors expand a little bit on the

motivation why it can be assumed that one and the same climbing fiber can be analyzed throughout sessions.

Our response: When we took images of climbing fibers, we also took images of brain surface vasculature and used them to locate the same field of views in subsequent imaging session. In addition, we always included mossy fibers in the granule cell layer (they also express EGFP as shown in Fig. 4A right) and Purkinje cell somata (in the case of double transgenic animals) in the z-stack images of climbing fibers. The spatial pattern of mossy fibers and; tdTomato expressing and non-expressing Purkinje cell somata were extremely stable and unique to each field of view. Therefore, we always had multiple references for finding the same climbing fibers: the morphology of the climbing fibers, the unique pattern of mossy fibers, and the Purkinje cell somata immediately below the climbing fibers. This ensured that the same climbing fibers were imaged across every session. We have added this information in the “*In vivo* imaging” subsection of the Methods (page 21, line 436-441).

Reviewer #2:

In this manuscript, the authors used *in vivo* time-lapse microscopy and mouse transgenic reporter lines to re-examine the behavior of climbing fiber collateral re-growth in the adult cerebellum. The technology is excellent and the area of research is interesting.

Although the authors present some intriguing data, I am confused about the interpretations of the results. My feeling is that this confusion arises because the authors would like to make a strong case that their results argue against the previous works from the Hawkes, Rossi, and Sherrard groups. But based on the data that is shown here, I am not so sure the results are quite so different from these earlier reports. But even if they are different as suggested, then a question has to be raised about the very different methodological approaches that were taken to disrupt the climbing fiber system.

Our response: We appreciate the reviewer’s positive comments on our technical approach and research topic, and are very thankful for the helpful critique that allowed us to significantly improve the clarity of our experimental model. As detailed below, we have tried to thoroughly address each comment and believe that the interpretations of the results is clearer in this revision.

Major criticisms:

1) This manuscript is driven by the argument that climbing fiber collateral sprouting after lesioning the olive does not selectively re-innervate the cerebellar zones that were targeted before lesion. However, I am concerned about how the zones are delineated, since the tdTomato expression (a proxy for ZebrinII expression) shows something peculiar. The zones defined by this transgenic are not so sharp. If this is really the case and not just an imaging artifact, then climbing fibers that are normally targeted to these "red" Purkinje cells would also sprout to them - in panel 5C you can see Red labeling in supposedly negative zones. That is, the boundaries between zones as not as sharp as one would expect of zebrinII expression. Therefore, it's not so surprising to me that you would get sprouting to both zones given that the boundaries, as defined here, are not so well defined in the first place.

Our response: To examine whether the expression pattern of tdTomato in our double transgenic mouse line matches with the endogenous zebrin II/aldolase C expression, we performed immunohistochemistry to label zebrin II/aldolase C in fixed cerebellar sections derived from our double transgenic mice. As shown in the new supplementary figure 3, the zonal boundary identified by tdTomato expression matches with the boundary identified by zebrin II/aldolase C expression. This is consistent with the previous study by Tsutsumi et al., J. Neurosci 35, 843-852 (2015), in which the authors confirmed identical zonal patterning of tdTomato and zebrin II/aldolase C expression in the same reporter mice. Furthermore, we have shown that the weak expression of tdTomato in the zone of tdTomato/zebrin II negative Purkinje cells is due to the expression of tdTomato in Bergmann glia (Supplementary Fig. 3B). This is consistent with the previous study by Fujita et al., PLOS ONE 9, e86679 (2014), in which the authors used a different line of reporter mice and showed that Bergmann glia express zebrin II/aldolase C. Importantly, glial expression of tdTomato does not interfere with our ability to unambiguously identify tdTomato positive and negative Purkinje cells *in vivo* (Fig. 4B). These data indicate that zebrin II/aldolase C positive and negative zones were accurately identified in our experiments.

2) I am cornered that the manipulation is not specific enough to make claims about zone specific ablation and regrowth. The authors mention that after 3-AP injection climbing fibers in both the positive and negative zebrinII zones (adjacent compartments) show some degree of climbing fiber degeneration, but I suppose the degeneration is more predominant in one zone over the neighboring one. Still, why then is it surprising that you would get regrowth into both compartments? Even if a single fiber can be tracked and is found to cross a boundary, I am not sure what this means since the whole compartmental integrity is compromised. That is, even if the authors are correct, its not so surprising that zones become irrelevant when there is massive destruction of the circuit.

Our response: We apologize that the original manuscript was confusing about our lesion model. We did not perform zone-specific ablation of climbing fibers. This is because our purpose was to examine whether newly sprouted climbing fiber collaterals, originating in any zone, cross the zonal boundary. Therefore, the lesion did not need to be specific to any particular zone. We made this point clear in this revision (page 9, line 182-183). Regarding the overall integrity of the olivocerebellar circuit in our lesion model, as shown in the new supplementary figure 1, only a small population of climbing fibers was ablated, suggesting that expansive destruction of the circuit did not occur. Nonetheless, at least in mice, newly sprouted climbing fiber collaterals ignored the zonal boundary. This is surprising because the previous study by Zagrebelsky et al., Eur. J. Neurosci. 8, 1051-1054 (1996) suggests that newly sprouted climbing fiber collaterals respect the zonal boundary even after near complete destruction of the olivocerebellar circuits (90-99 % of climbing fibers were ablated).

3) Based on the two points above, and the fact that the methodology used in this study is completely different from the other groups (and others used rat), I am not sure such strong comparisons can be made. In fact, the design of the study here seems like this paper would address a totally different set of questions to me. In end, this paper likely tells us more about competitive rewiring of a general territory when innervation to multiple compartments is disrupted rather than a zonal specific story.

Our response: The previous study mentioned in this comment (Zagrebelsky et al., 1996) used systemic injection of 3-AP for ablating climbing fibers in rats to study the interaction between zebrin II zones and climbing fiber collateral sprouting. The same group thoroughly characterized climbing fiber collateral sprouting in their lesion model in Rossi et al., J. Comp. Neurol. 308, 513-535, 536-554 (1991). Characteristics of collaterals sprouting in our present study are very similar to those reported in Rossi et al (1991). First, the morphology of surviving climbing fibers after lesion is almost identical between the two studies. Second, surviving climbing fibers start sprouting new collaterals within a week after the lesion (Fig. 1D). Third, newly formed climbing fiber ladders apparently form functional synapses (Fig. 1E and Supplementary Fig. 4). Fourth, post-lesion climbing fiber growth continues at least for several months (Fig. 2 and 5). These considerable similarities indicate that, although we used local injection of 3-AP in mice, we observed the same form of post-lesion climbing fiber collateral sprouting as Rossi et al (1991) and Zagrebelsky et al (1996).

Furthermore, we now present new data indicating that tdTomato expression accurately identifies the zonal boundary (Supplementary Fig. 3) and that our lesion selectively affects a small population of climbing fibers in the cerebellar cortex (Supplementary Fig. 1). These new data support our claim that we addressed how the zonal boundary affects post-lesion climbing fiber collateral sprouting, and thus, comparing our results with Zagrebelsky et al (1996) is reasonable. But, we agree that the differences in lesion models used in our study and Zagrebelsky et al (1996) should be clearly mentioned. Therefore, in this revision, we made the differences (mouse vs. rat, and local vs. systemic injection) clear in the discussion section (page 15, line 321-326).

4) In the discussion, the authors spend a whole paragraph discussing the BDNF work from Sherrard, almost in support of their data. This is confusing because Dr. Sherrard made the claim for zonal re-growth. But, Nishiyama and colleagues suggest that even though non-native zones are re-innervated, there is still a preference for the native zone. As I mentioned above, is there really any major difference between what they see and what was previously reported? Point number (3) at the end of the first paragraph in the Discussion section basically hints at similar outcomes. Even if the collaterals tend to go to a nearby zone, ultimately they still have a preferential target, even if proximity is the guiding factor. Thus, the take home message is confusing because a solid interpretation cannot be made.

Our response: We apologize that the discussion in the original manuscript was unclear about how our results relate to the studies by Sherrard and colleagues. Transcommisural climbing fiber re-innervation, studied by Sherrard and colleagues, is observed only in developing animals, whereas collateral sprouting in the present study was observed in adult animals. Therefore, we did not intend to compare those two different forms of post-lesion climbing fiber collateral sprouting. In this revision, we made these points clear in the discussion section (page 16, line 347-350). But, we still mention transcommisural climbing fiber re-innervation because it provides an important general insight into the functional significance of newly sprouted climbing fiber collaterals.

5) Injecting 3-AP directly into the olive seems like a major perturbation. How was the spread of the toxin controlled? I would assume that this method kills more than just the native and non-native adjacent zones that are described. The authors need to provide some detailed histological

characterization of what these injections do - for example, cell death in the olive and also within cerebellar cortex, low power images of the injection spot in the olive to show the damage done because of the toxin and also because of physical/mechanical damage from the injection itself. What about adjacent nuclei such as the lateral reticular nuclei? Are they damaged too? Do they re-grow? Some of the profiles in panel 5C look like ectopic mossy fibers in the molecular layer (look at the fiber and terminal just to the left of the 6 in the label "6weeks"). Perhaps some of the abnormal growth that is seen is from mossy fiber collateral regrowth? 200 nanoliter injection is quite large, so a clear and thorough analysis of regional pathology should be provided.

Our response: To address this comment, we present our histological characterization of the cerebellar cortex and the medulla (new supplementary figure 1). Lesion in the cerebellar cortex was examined by immunolabeling of vesicular glutamate transporter 2, which labels all climbing fiber and mossy fiber terminals (Supplementary Fig. 1A and 1B). Lesion in the medulla was examined by fluorescent Nissl staining (Supplementary Fig. 1C-1F). The tissue damage in the medulla appeared not to be perfectly specific yet was highly localized to the inferior olive (Supplementary Fig. 1F). Importantly, mossy fiber terminals appeared unaffected throughout the cerebellar cortex (Supplementary Fig. 1B1), including the regions near the degenerated climbing fibers (Supplementary Fig. 1B2 and 1B3). These results indicate that damage is specific to the olivocerebellar circuit in our lesion model.

A subpopulation of mossy fibers express EGFP in our reporter mice (Fig. 4A right), but EGFP-positive mossy fibers were found only in the granule cell layer. We have never observed ectopic mossy fibers in the molecular layer of the reporter mice. This was also the case after lesioning. New figure 1E and supplementary figure 4 show that EGFP-labeled structures in the molecular layer are only climbing fibers; no ectopic mossy fibers were found even after lesioning.

It should be noted that all of our climbing fiber images taken *in vivo* are presented as maximum projections showing a top-down view. This is a standard way of showing *in vivo* images, which is used in almost all similar *in vivo* imaging publications. In a maximum projection, a segment of climbing fiber ladder that runs mostly parallel to the optical axis (z-axis) may be seen as a large axon terminal such as a mossy fiber terminal. This is because fluorescent signals derived from different z-planes of such a climbing fiber segment (i.e., a segment that runs mostly parallel to the optical axis) are all projected onto similar locations in a single x-y plane. In this revision, we made it clear in figure legends that maximum projections are used to show *in vivo* images of climbing fibers.

Additional note: As we described in the “Image analysis and statistics” subsection of the Methods (page 21, line 447), we performed quantitative analyses on 3-dimensional z-stacks, not 2-dimensional projections. Therefore, projection did not affect our analyses.

Additional concerns:

1) The title of the paper is much too broad. There is no mention of cerebellum, climbing fiber, zones, module, etc. I am also not sure what the authors mean by "axonal remodeling".

Our response: We changed “axonal remodeling” and “brain” in the original title to “axonal sprouting” and “cerebellum”, respectively.

2) In the figure panels, the layers of the cerebellar cortex should be labeled. This will give readers that are not in the cerebellar field some context to work with.

Our response: We added the layers of the cerebellar cortex in the images in which multiple layers are clearly shown (Fig. 4 and Supplementary Fig. 1, 3, 4). But, only the molecular layer is shown in *in vivo* images, except for figure 4B. Figure 1B and 1E (images of cerebellar sections) show mostly molecular layer. For those images showing a single layer (molecular layer), we did not label the layer, but clearly mentioned in the figure legends that images were taken from the molecular layer.

3) Along the same lines as above, even though it is mentioned in the methods which lobules are targeted, the lobule numbers should also be labeled on the images.

Our response: We added the lobule numbers in figure 4A and supplementary figure 1A. But, for *in vivo* images, we would prefer to mention the lobule numbers in the main text and figure legends. When we took *in vivo* images from the EGFP single transgenic mice, we placed a cranial window on lobule VI and VII. We then randomly imaged surviving climbing fibers in a field of view and did not try to identify whether the climbing fibers were located in lobule VI or VII. Except for a few instances in which a cranial window happened to be small and covered only lobule VII (Fig. 1A and 1D), we cannot tell confidently whether each climbing fiber was located in lobule VI or VII. On the other hand, all *in vivo* images were taken from lobule VIII in the EGFP and tdTomato double transgenic mice. However, it might seem strange if we label the lobule numbers only for the images taken from double transgenic mice and the labels are all lobule VIII. Additionally, no lobule specific differences were found; the pattern of climbing fiber sprouting was the same between the CFs imaged from lobule VI/VII (single transgenic mouse) and lobule VIII (double transgenic mouse) (Fig. 5A and 5B). It is our belief that the imaged lobule is now sufficiently clear in this revision, because we have mentioned the lobule numbers in the Results (page 6, line 112 and page 8, line 177) and added the information in the figure legends.

4) It is incorrect to refer to cerebellar zones as "slabs".

Our response: We replaced "slabs" to "bands" (page 4, line 72 and page 34, line 655).

5) The number of animals used for each experiment should be mentioned in the main body of the text.

Our response: We added the number of animals used for each experiment in the main text (page 6, line 113; page 8, line 178; and page 9, line 192).

Reviewer #3:

Axonal remodeling is a recognized cause and consequence of many neurological disorders. It may contribute to the formation of aberrant connections that elicit pathological network activity or to (partial) re-innervation of synaptic targets. In this study, Dhar and colleagues study lesion-induced remodeling of cerebellar climbing fibers (CFs) using an elegant *in vivo* approach that

relies on 2-photon time-lapse imaging of genetically labeled CFs following 3-AP injections into the inferior olive. The authors provide a very detailed description of how CF sprouting occurs, both temporally and spatially. This work provides several anatomical insights such as that sprouting CFs innervate relevant and non-relevant Purkinje cells (PCs) and that they do not obey to functional boundaries such as those marked by zebrin II.

This is a carefully conducted study that confirms and expands previous (immuno)histological data on CF sprouting. The technical approach used is elegant and has a lot of potential for the future dissection of mechanisms underlying CF sprouting. Unfortunately, the current study is purely descriptive and goes into excessive detail on how CF sprouting occurs. However, in my opinion no major insights into CF sprouting are presented. Such insight would require combining the presented tools (e.g. Nefl-eGFP mice, 2-photon imaging) with manipulation or visualization of specific cellular processes or molecules relevant to axonal remodeling.

Our response: We are thankful for the reviewer's highly positive comments on our technical approach, including its potential for future studies. The specific comments were also very helpful for improving the manuscript.

As the reviewer mentioned, the present study is based on observation of post-lesion axonal sprouting, rather than manipulation of the process or relevant molecules. However, we would like to claim respectfully that the present study still addresses an important question: do newly sprouted axon collaterals preferentially innervate functionally relevant targets in the adult brain? To address this question, we used long-term *in vivo* time-lapse microscopy to simultaneously visualize post-lesion axonal sprouting and functional structure of the mammalian brain for the first time. This approach has provided novel insight into how post-lesion sprouting of climbing fibers affects the functional structure of the cerebellum. Thus, we believe that the present study makes a major contribution in the field and we can reasonably wait for subsequent studies for manipulation of the sprouting process or relevant molecules.

1) Figure 2: I would suggest the authors to add more examples of CF sprouting. The graph in Figure 2B summarizes a lot of different examples of CF sprouting but it is unclear how similar or different the spatial patterns of CF remodeling can be. Do CFs always generate the same number of ladders?

Our response: We agree with the reviewer that we should show more examples, because the spatiotemporal pattern of collateral sprouting is not identical across different climbing fibers. In this revision, we provide more examples of climbing fiber images and traces in new supplementary figure 2.

2) Throughout the manuscript, the authors state that CF collaterals generate new synaptic connections. I believe this idea is based on the characteristic morphology of the collaterals? However, how do the authors know that synapses are formed and are these different for relevant and non-relevant PCs?

Our response: In the original manuscript, we assumed that newly formed climbing fiber collaterals form synapses based on the previous electron microscopic analysis by Rossi et al., J. Comp. Neurol. 308, 536-554 (1991). Rossi et al (1991) found that, after systemic injection of 3-

AP in rats, newly sprouted climbing fiber collaterals formed typical asymmetric synapses with Purkinje cells with morphology that is characteristic of this synapse in a normal cerebellum (this study is cited earlier in this revision, page 5, line 109-111). Although we used a slightly different lesion model in the present study (local rather than systemic injection of 3-AP in mice), many characteristics of post-lesion climbing fiber collateral sprouting are almost identical between the two studies. This strongly suggested that new climbing fiber collaterals form synapses in our lesion model.

Furthermore, we provide new data in this revision that support synapse formation by newly sprouted climbing fiber collaterals. Vesicular glutamate transporter 2 (VGLUT2) is a well-established marker for presynaptic site of climbing fibers. We performed immunolabeling of VGLUT2 two weeks after lesioning and found that newly formed climbing fiber ladders were all VGLUT2-positive (Fig. 1E) regardless of whether they appeared in native or non-native zones (Supplementary Fig. 4). These results indicate that newly formed climbing fiber ladders form synapses regardless of whether they innervate relevant or non-relevant Purkinje cells.

3) Perhaps the authors can add additional labels to Figures 5A and 5B to emphasize the difference between these two graphs (inside vs outside addition).

Our response: We added the labels in the revised Figure 5A and 5B.

REVIEWERS' COMMENTS:

Reviewer #1 (Remarks to the Author):

The authors have met my comment in a satisfactory fashion.

Reviewer #2 (Remarks to the Author):

The authors have provided a thoroughly revised version of the manuscript and have provided a point by point response to all the comments. I am satisfied with the changes and the explanations that they have provided.

Reviewer #3 (Remarks to the Author):

Dhar and colleagues have addressed several of my initial concerns by including data on the localization of synaptic structures, more examples of CF sprouting and by re-labeling of Figures. My main concern with this study was and is that it is purely descriptive; it uses nice technology to study anatomical characteristics of axon regeneration but given the data previously available the authors should have used their tools and approaches to address more mechanistic questions relevant to the field. As indicated previously, I do not feel that in its present form this study should be published in a high-impact, multi-disciplinary journal.

Point-by-point response to reviewers for Dhar et al., NCOMMS-16-01043A

The reviewers' comments on our revised manuscript are written in black and our responses are written in blue below.

Reviewer #1:

The authors have met my comment in a satisfactory fashion.

Our response: We are pleased to hear that the reviewer is satisfied by our revised manuscript.

Reviewer #2:

The authors have provided a thoroughly revised version of the manuscript and have provided a point by point response to all the comments. I am satisfied with the changes and the explanations that they have provided.

Our response: We are pleased to hear that the reviewer is satisfied by our revised manuscript.

Reviewer #3:

Dhar and colleagues have addressed several of my initial concerns by including data on the localization of synaptic structures, more examples of CF sprouting and by re-labeling of Figures. My main concern with this study was and is that it is purely descriptive; it uses nice technology to study anatomical characteristics of axon regeneration but given the data previously available the authors should have used their tools and approaches to address more mechanistic questions relevant to the field. As indicated previously, I do not feel that in its present form this study should be published in a high-impact, multi-disciplinary journal.

Our response: Although we appreciate the reviewer's feedback on our original and revised manuscripts, we would like to argue against the comment that our work is purely descriptive and thus not suitable for a high-impact, multi-disciplinary journal. Before this study, it was unclear how post-lesion axonal reinnervation proceeds in relation to the existing functional circuits of the brain. This is an important question especially because of the potential role of axonal reinnervation in functional recovery after brain damage. We have addressed this question by characterizing the precise spatiotemporal pattern of lesion-induced axonal sprouting in a defined functional circuit of the mammalian brain for the first time *in vivo*. Our results suggest that existing functional circuits do not limit the sprouting of axon collaterals, but that the collaterals continue to have their greatest synaptic influence near their original targets. Therefore, the integrity of functional circuits might be loosely maintained in brain areas in which functionally relevant neurons are clustered together. In our view, these novel and fundamental findings should be of interest to not only neuroscientists but also biologists interested in translational research as well as clinicians.